# Cystic Fibrosis Newborn Screening: A Systematic Review-Driven Consensus Guideline from the United States Cystic Fibrosis Foundation

**DOI:** 10.3390/ijns11020024

**Published:** 2025-04-02

**Authors:** Meghan E. McGarry, Karen S. Raraigh, Philip Farrell, Faith Shropshire, Karey Padding, Cambrey White, M. Christine Dorley, Steven Hicks, Clement L. Ren, Kathryn Tullis, Debra Freedenberg, Q. Eileen Wafford, Sarah E. Hempstead, Marissa A. Taylor, Albert Faro, Marci K. Sontag, Susanna A. McColley

**Affiliations:** 1Department of Pediatrics, University of Washington School of Medicine, Seattle, WA 98105, USA; meghan.mcgarry@seattlechildrens.org; 2Center for Respiratory Biology and Therapeutics, Seattle Children’s Research Institute, Seattle, WA 98101, USA; 3Department of Genetic Medicine, Johns Hopkins University School of Medicine, Baltimore, MD 21205, USA; 4Departments of Pediatrics and Population Health Sciences, University of Wisconsin School of Medicine and Public Health, Madison, WI 53792, USA; pmfarrell@wisc.edu; 5Community Representative to the CF Foundation, Bethesda, MD 20814, USA; faith.clark85@yahoo.com (F.S.); kpadding@cff.org (K.P.); cwhite@cff.org (C.W.); 6The Cystic Fibrosis Foundation, Bethesda, MD 20814, USA; shempstead@cff.org (S.E.H.); mataylor@cff.org (M.A.T.); afaro@cff.org (A.F.); 7Laboratories Administration, Maryland Department of Health, Baltimore, MD 21205, USA; mary.dorley@maryland.gov; 8Department of Pediatrics, The Pennsylvania State College of Medicine, Hershey, PA 17033, USA; shicks1@pennstatehealth.psu.edu; 9Department of Pediatrics, Children’s Hospital of Philadelphia, Philadelphia, PA 19104, USA; renc@chop.edu; 10Department of Pediatrics, Nemours Children’s Hospital, Wilmington, DE 19803, USA; kathryn.tullis@nemours.org; 11Division of Newborn Screening and Genetics, Texas Department of State Health Services, Austin, TX 78714, USA; dfreedenberg@geneaustin.org; 12Medical Geneticist and Consultant to the CF Foundation, Bethesda, MD 20814, USA; 13Galter Health Sciences Library, Northwestern University Feinberg School of Medicine, Chicago, IL 60611, USA; q-wafford@northwestern.edu; 14Center for Public Health Innovation, Evergreen, CO 80439, USA; marci.sontag@cphinnovation.org (M.K.S.); smccolley@luriechildrens.org (S.A.M.); 15Stanley Manne Children’s Research Institute, Ann & Robert H. Lurie Children’s Hospital of Chicago, Chicago, IL 60611, USA

**Keywords:** cystic fibrosis, newborn screening, immunoreactive trypsinogen, cystic fibrosis transmembrane conductance regulator, F508del (p.Phe508del), genetic testing

## Abstract

Newborn screening for cystic fibrosis (CF) has been universal in the US since 2010; however, there is significant variation among newborn screening algorithms. Systematic reviews were used to develop seven recommendations for newborn screening program practices to improve timeliness, sensitivity, and equity in diagnosing infants with CF: (1) The CF Foundation recommends the use of a floating immunoreactive trypsinogen (IRT) cutoff over a fixed IRT cutoff; (2) The CF Foundation recommends using a very high IRT referral strategy in CF newborn screening programs whose variant panel does not include all CF-causing variants in CFTR2 or does not have a variant panel that achieves at least 95% sensitivity in all ancestral groups within the state; (3) The CF Foundation recommends that CF newborn screening algorithms should not limit *CFTR* variant detection to the F508del variant or variants included in the American College of Medical Genetics-23 panel; (4) The CF Foundation recommends that CF newborn screening programs screen for all CF-causing *CFTR* variants in CFTR2; (5) The CF Foundation recommends conducting *CFTR* variant screening twice weekly or more frequently as resources allow; (6) The CF Foundation recommends the inclusion of a *CFTR* sequencing tier following IRT and *CFTR* variant panel testing to improve the specificity and positive predictive value of CF newborn screening; (7) The CF Foundation recommends that both the primary care provider and the CF specialist be notified of abnormal newborn screening results. Through implementation, it is anticipated that these recommendations will result in improved sensitivity, equity, and timeliness of CF newborn screening, leading to improved health outcomes for all individuals diagnosed with CF following newborn screening and a decreased burden on families.

## 1. Introduction

### 1.1. Historical Perspective and Terminology

Newborn screening (NBS) for cystic fibrosis (CF) became feasible in 1979 [1], measuring immunoreactive trypsinogen (IRT) using a dried blood spot specimen. Infants whose first IRT was out of range (elevated) had a second IRT measurement performed, termed an IRT-IRT protocol [2,3]. In the early 1990s, following the identification of the cystic fibrosis transmembrane conductance regulator (*CFTR*) gene and the common c.1521_1523del (p.Phe508del; legacy: F508del or [delta]F508) variant, some NBS programs replaced the second IRT measurement with molecular testing for F508del, introducing the IRT-DNA protocol for CF NBS [4,5,6]. This protocol allowed programs to lower the IRT cutoff, which yielded higher sensitivity and faster identification of infants with CF, though diagnoses were biased toward infants of European descent who were more likely to harbor the F508del variant. Over the next 20 years, advances in genetic testing technology, identification of more *CFTR* variants, and improved understanding of variant pathogenicity and genotype–phenotype correlation led to expanded multi-variant panel analysis being the preferred IRT-DNA strategy [7] in the US. Critical to these advances has been the Clinical and Functional TRanslation of CFTR (CFTR2) project, which began in 2012 with the goal to describe genetic variants observed in individuals with CF and to provide variant interpretation along with clinical correlation via a publicly available website (https://cftr2.org [accessed on 5 February 2025]) [8]. As of September 2024, 1167 *CFTR* variants had been interpreted by CFTR2, with 1085 deemed CF-causing, e.g., these variants are expected to result in CF when found opposite (in *trans* with) another CF-causing variant. Variant interpretation by CFTR2 has facilitated laboratories in identifying and interpreting *CFTR* variants to include in CF NBS and for carrier and diagnostic testing purposes. More recently, next-generation sequencing (NGS) has been utilized, which is a technique that allows faster interrogation of all or part of *CFTR* at higher volumes. Coupled with expanded knowledge of disease propensity of *CFTR* variants provided by CFTR2, NGS has recognized utility as part of CF NBS. Sequencing results reported following variant panel analysis result in an algorithm termed IRT-DNA-SEQ (also referred to as IRT-DNA-EGA [extended gene analysis]) [9]. Reporting software can be used to limit sequencing results to a customizable list of hundreds of variants that constitute a large panel for states choosing to continue with an IRT-DNA protocol for CF NBS [10].

Over time, technological advances and the resulting options for CF NBS algorithms, the development of multiple variant interpretation schemes, and increased categories of infants identified by CF NBS have necessitated clear definitions such that programs can make informed choices. Definitions of the terminology used in and selected for this manuscript can be found in Table 1.

### 1.2. Disparities in CF Newborn Screening and Diagnosis

Newborn screening for CF is performed in many countries worldwide and has been performed in all US states since 2010 [14], where it falls under the purview of state public health departments. Variations among states in the US regarding CF NBS algorithms and follow-up practices can lead to differences in equity, sensitivity, and timeliness of intervention [14]. Individuals from historically marginalized races or ethnicities (social constructs that typically reflect physical characteristics and shared culture, and which may be tied to a person’s genetic lineage or ancestry) are disproportionately impacted by variations in screening practices, resulting in a screening paradigm that does not equitably identify infants at risk for CF [15,16]. This was a concern raised as CF NBS was being implemented [17], as it was recognized early that the routine *CFTR* panels used in CF NBS, frequently including the 23 variants recommended for use in CF carrier screening by the American College of Medical Genetics and Genomics (termed the ACMG-23) [18,19], favored the identification of non-Hispanic white infants with CF.

Implementation of CF NBS is associated with improved nutritional outcomes and better lung function, even when accounting for the improvements in these measures that have resulted from advances in CF clinical care over time [20,21]. Infants not identified by NBS or who have a delayed presentation to a CF specialist have poorer clinical outcomes than those with an early presentation [20,22,23]. Earlier age at sweat test, clinical evaluation, or hospitalization at a CF Center is associated with better outcomes, such as improved growth [22]. Furthermore, delays in diagnosis or clinical care due to inequities in the screening algorithms or misconceptions of risks contribute to existing health disparities among Black and Hispanic populations with CF [24,25,26,27,28,29,30]. Misconceptions of risks that lead to diagnostic delays may be related to the prevalence of CF in Black and Hispanic populations (lower than in non-Hispanic white populations), including erroneous beliefs that the disease does not occur in these populations, or related to the conflation of a screening vs. diagnostic test [31]. In the case of CF NBS, some providers may be overly confident in dismissing CF in the setting of a normal CF NBS result, believing that the test performed was diagnostic and that a normal result eliminates CF as a possibility. Indeed, CF NBS is a screening test and should be viewed as identifying infants at higher risk for CF, but not definitively ruling out CF in infants who are not detected.

### 1.3. Rationale for the CF Newborn Screening Guideline

Despite many reports on the outcomes of NBS at the level of individual programs or countries, there are currently no consensus recommendations to guide best practices for CF NBS in the US. While most NBS programs aim to maximize the detection of CF-affected infants and thereby minimize missed cases, an acceptable threshold for sensitivity has not been established. A reasonable goal for NBS programs is a 95% sensitivity or true-positive detection rate for all infants with CF, with equitable ascertainment within major ancestral sub-populations based on known *CFTR* variants in those populations. This threshold has precedence from (1) the European CF Society guidelines for CF NBS, though without specification that infants from all ancestral backgrounds be equitably detected [32], and (2) the American College of Medical Genetics and Genomics recommendations for CF carrier screening, which specifies that 95% detection must be achieved within the major ancestral groups in the US as defined by the US Census [33]. In addition to IRT and molecular testing, we address testing frequency and communication strategies, both of which should be considered in NBS best practices.

This report is a product of the CFF newborn screening initiative, which arose with the recognition of an increasingly diverse CF population [34], CF NBS, and the convergence of CFTR2-driven expanding variant pathogenicity information and sequencing technology advances. The goal of the initiative was to improve equity and reduce disparities in diagnosis and outcomes across races, ethnicities, and ancestral groups. This guideline aims to describe NBS program practices that improve equity, sensitivity, and timeliness in diagnosing infants with CF. Through implementation, we anticipate improved health outcomes for all individuals with CF diagnosed following NBS in the US and globally.

## 2. Methods

### 2.1. Cystic Fibrosis Foundation Committee

The US CF Foundation convened a multidisciplinary committee of CF clinicians and researchers, parents of children with CF who had NBS, state NBS program experts, a primary care pediatrician, and a genetic counselor from the US. The committee collaborated to develop six systematic reviews. Each systematic review and protocol was registered in PROSPERO [35], prior to the commencement of the literature review (PROSPERO Registration numbers: CRD42023481926, CRD42024500201, CRD42024522404, CRD42024535075, CRD42024542279, CRD42024552545). Methodological details are provided in the Appendix A.

### 2.2. PICO Strategy, Application and Timeline

Population, Intervention, Comparison, and Outcome (PICO) questions were developed to identify the NBS processes that impact equity, sensitivity, and timeliness of diagnosing infants with CF. Cost-effectiveness, resource allocation, and implementation were not addressed, though these factors are recognized as having a significant impact on the feasibility of implementing these consensus recommendations. The identification of infants with *CFTR*-related metabolic syndrome (CRMS) or CF Screen Positive, Inconclusive Diagnosis (CFSPID), as well as the diagnosis and care of infants with CF and genetic counseling, were not addressed, as there are prior guidelines on each of these topics from the CFF or other groups [13,36,37,38]. The six systematic reviews were conducted between November 2023 and May 2024. The voting committee of fourteen people, including five CF clinicians and researchers, three parents of children with CF, three state NBS program experts, a primary care pediatrician, a CFF staff member, and a genetic counselor, convened to vote in person on recommendation statements in July 2024. An 80% consensus was the a priori threshold for approval.

The CFF intends for these consensus recommendations to provide evidence-based guidance to clinicians, patients, NBS programs, pediatricians, and other stakeholders. These consensus recommendations are intended to be location-agnostic and apply to newborns screened for CF throughout the US. It is recognized that full implementation of the guideline may be aspirational for some programs because of the significant changes to systems, processes, and allocation of resources that would be required. However, even incremental changes implemented as resources allow will contribute to progress toward a more sensitive, equitable, and timely CF NBS process.

### 2.3. CFTR Variant Interpretation

The terminology for the interpretation of the clinical significance of *CFTR* variants is not uniform across NBS laboratories, other diagnostic testing labs, or expert groups. In this manuscript, we have used the term CF-causing [8] to define *CFTR* variants that should be reported as part of NBS following panel testing or sequencing (Table 1). Recognizing that not all CF-causing variants are readily detectable by sequencing technology, NBS programs should focus on CF-causing variants interpreted as such by CFTR2 and which are located in the coding and flanking regions of *CFTR*. Programs may further choose whether to include other types of variants as part of their NBS algorithms, such as structural variants, deep intronic variants, variants that may not always lead to CF, and variants not interpreted by CFTR2.

## 3. Results

The CF NBS committee developed seven recommendation statements (Table 2). Based on the quality and limited number of studies, the recommendations represent the committee’s consensus opinion. All statements reached 100% agreement.

### 3.1. Immunoreactive Trypsinogen

All CF NBS protocols begin with a newborn bloodspot measurement of immunoreactive trypsinogen (IRT) (Figure 1) [39], which is typically elevated in infants with CF but may also be elevated in CF carriers or for other reasons, including ambient temperature and seasonality, kit-to-kit variations, infant overall health and nutritional needs, gestational age, and ancestry [40,41,42,43,44,45,46]. Some NBS programs perform two IRT measurements, either on all infants or only for those in whom the first measurement is elevated. In all US programs, infants determined as having an elevated IRT by their CF NBS protocol have their sample reflexed to the next CF NBS tier, which is the molecular testing of *CFTR* variants.


**Recommendation** **1:**

**The Cystic Fibrosis Foundation recommends the use of a floating immunoreactive trypsinogen cutoff over a fixed immunoreactive trypsinogen cutoff.**


Infants with CF who are missed by NBS frequently have an IRT value below the threshold for detection in the first tier of the NBS algorithm [9,47,48,49]. Investigations into CF cases missed by NBS in Colorado found that seven of eight cases missed using a fixed cutoff of 60 ng/mL would have been identified using a floating cutoff of the 96th percentile [50]. A survey of 13 state NBS labs determined that missed cases occurred more frequently in states with a fixed cutoff than those with a floating cutoff (3.48% vs. 2.36%, respectively) [30]. Using a floating cutoff is expected to minimize environmental impacts on IRT, such as changes in temperature throughout the seasons and kit-to-kit or lot variations, resulting in fewer missed cases [51]. However, a floating cutoff will not adjust for infant-specific circumstances, such as birth stress.

Though the evidence reviewed supported a recommendation for a floating IRT cutoff over a fixed one, the sensitivity and specificity of detecting infants with CF will also be impacted by the specific thresholds used. Fixed cutoffs among states vary from 50 ng/mL to 100 ng/mL, and floating cutoffs vary from the 95th to the 98.8th percentile [14]; such variation contributes to variability in sensitivity. Early review of the Colorado CF NBS program in the 1980s resulted in lowering the fixed IRT threshold, albeit for the second IRT in the state’s IRT-IRT protocol, from 120 ng/mL to 80 ng/mL due to a missed case [2]. Conversely, other studies did *not* find a significant effect on sensitivity due to changing either a floating or fixed IRT threshold, though other outcomes such as specificity, positive predictive value, and number of sweat test referrals *were* impacted by these changes [46,52,53]. Of note, some NBS programs using a floating IRT cutoff have a backup fixed IRT cutoff to ensure the floating cutoff does not drift too high [50]; others have explored modeling a fixed cutoff that may be adjusted seasonally [46].

The recommendations apply to the IRT measurement on the initial sample and do not address procedures for an IRT measurement on a second sample when used as a part of an NBS protocol (IRT-IRT-DNA). Programs that routinely collect a second sample may consider employing a similar approach after careful examination of the IRT values from the second screen.


**Recommendation** **2:**

**The Cystic Fibrosis Foundation recommends using a very high immunoreactive trypsinogen referral strategy in CF newborn screening programs whose variant panel does not include all CF-causing variants in CFTR2 or does not have a variant panel that achieves at least 95% sensitivity in all ancestral groups within the state.**


The very high IRT (VHIRT) strategy, sometimes called “failsafe” or “safety net”, involves the designation of infants as screen-positive if the IRT measurement is above a particularly high threshold (typically greater than the 99th percentile) and when no *CFTR* variants are detected by the second-tier variant panel. Use of the VHIRT strategy began when the second-tier variant panels included only F508del, or a small number of variants, to increase the likelihood that infants with CF whose *CFTR* variants were not included in the panel had positive NBS tests [7,54]. The VHIRT strategy results in many referrals but few diagnoses, leading to questioning its value or elimination from the CF NBS algorithm [9,54,55].

Evidence supports that a VHIRT strategy successfully identifies infants whom CF NBS would have otherwise missed with a limited variant panel. From 1991 to 2003, 18 out of 227 (8%) infants with CF from Victoria, Australia were diagnosed following VHIRT referral when only F508del was assessed [56]. From 2002 to 2005, New York reported that five out of 106 (5%) infants with CF were diagnosed following VHIRT referral after a 32-variant panel [57]. A retrospective analysis of a newly adopted NBS algorithm in Colorado indicated that VHIRT would have identified two of three missed CF cases between 2006 and 2008 when a 43-variant panel was utilized [58]. The VHIRT strategy predominantly benefits infants of ancestral groups that frequently have less commonly detected *CFTR* variants. For example, among 33 cases of CF, CRMS/CFSPID, and possible CF/CRMS/CFSPID identified by VHIRT in New York from 2003 to 2013, 75% were Black, Hispanic, Asian, American Indian, or multiracial [59].

The impact of VHIRT is lessened with expanded *CFTR* panels used for CF NBS. Given that CFTR2 contains 1167 interpreted variants at the time of this report (https://cftr2.org [accessed on 5 February 2025]), accounting for >98% of alleles reported in the CFTR2 dataset, committee consensus was that VHIRT is not necessary if the second-tier panel either uses all CF-causing variants in CFTR2 or is of adequate size and makeup to detect ≥95% of the variants within all ancestral groups of the screened populations.

### 3.2. Molecular Testing of CFTR

For infants whose bloodspot sample progresses to the second tier of the NBS program for molecular analysis, a pre-determined number of *CFTR* variants are assessed and reported as part of a *CFTR* variant panel (Figure 1). Notably, at this step, sequencing methodology may be used but is still considered a *CFTR* variant panel when only a limited and pre-determined number of *CFTR* variants are reported.


**Recommendation** **3:**

**The Cystic Fibrosis Foundation recommends that CF newborn screening algorithms should not limit *CFTR* variant detection to the F508del variant or variants included in the ACMG-23 panel.**


In 2001, followed by a revision in 2004, the ACMG recommended a panel of 23 pathogenic *CFTR* variants, including F508del, for use in prenatal CF carrier screening [18,19]. Even though the variants included on the panel were derived primarily from CF populations with Northern European or Ashkenazi Jewish ancestry, many CF NBS programs eventually used either F508del alone or the limited ACMG-23 panel to detect CF in the entire population undergoing NBS [33]. However, as of 2023 and in response to calls for equity in *CFTR* variant detection, ACMG no longer recommends the 23-variant panel, instead replacing it with a minimum of 100 variants [33] to achieve better and more equitable CF carrier detection among major US ancestral groups.

Limiting the screening variant panel to only F508del or the ACMG-23 panel has relatively low sensitivity in the overall CF population; sensitivity ranged from 56% to 90% with F508del only [7,16,60,61,62,63,64] and 47 to 93% with the limited ACMG-23 panel [16,49,61,62]. Sensitivity with screening for F508del or the ACMG-23 panel is even lower in people who are American Indian, Asian, Black, Hispanic, or multiracial than in the overall CF population, resulting in an unacceptably high rate of false-negative NBS results [16,49,61,62]. These missed infants face significant barriers to diagnosis due to the misperception that CF does not occur in all populations. In the United States, there is strong evidence of racial and ethnic inequities in pediatric health care [65]. Furthermore, the “normal” NBS result itself can be a barrier to further testing in all populations [66,67,68].


**Recommendation** **4:**

**The Cystic Fibrosis Foundation recommends that CF newborn screening programs screen for all CF-causing *CFTR* variants in CFTR2.**


Several NBS panels include an extended number of variants (i.e., more than 100), resulting in improved sensitivity compared to limited variant panels [16,48,62]. However, even when screening is performed using an extended *CFTR* variant panel, the sensitivity of detecting one variant is less than 95% and is much lower in American Indian, Asian, Black, Hispanic, or multiracial populations [16,48,62]. Screening for all *CFTR* variants that have been interpreted by CFTR2 and which are deemed CF-causing (Table 1) improves sensitivity in the overall CF population compared to screening with extended variant panels [16,48,49]. This increase in sensitivity is particularly improved in the American Indian, Asian, Black, Hispanic, or multiracial populations [49].

Improving sensitivity requires not only increasing the number of *CFTR* variants screened, but also screening for the *CFTR* variants that occur more frequently in American Indian, Asian, Black, Hispanic, or multiracial populations [48,69,70]. This can be achieved by screening for all CF-causing variants in CFTR2 but may also be achieved if specific targeted variants from ancestral groups in each jurisdiction are added to an existing screen. This improves equity in diagnosis, decreases the time to diagnosis [9], and offers the best option to ensure NBS for CF is a public health benefit for all infants [71]. Because the ancestral backgrounds of populations change over time, this strategy requires periodic reassessment of the population of the region.

Notably, the guideline committee decided to use the CFTR2-defined interpretation of CF-causing to determine the variants that should be assessed by NBS (Table 1). It is recognized that screening for structural variants (large exon deletions or duplications) and deep intronic variants requires significant changes to the components, design, and analytics of the test. Programs should test for all CF-causing variants in the coding and flanking regions of *CFTR* and consider deletion/duplication analysis and deep intronic variant screening when feasible. In addition to the inclusion of CF-causing variants, CF NBS programs may also choose to include variants of varying clinical consequences (VVCC) based on their program goals related to the detection of CRMS/CFSPID. Accordingly, programs may also choose whether to refer only infants with two *CFTR* variants for diagnostic follow-up and sweat testing after screening for CF-causing variants in CFTR2 or refer infants with one or two variants for NBS follow-up (Figure 2; see also *Considerations for Implementation*). Programs referring only infants with two CF-causing variants should continue to notify parents of presumed carriers (only one CF-causing variant was identified) and recommend that they receive genetic counseling, even though infants would not be referred for diagnostic follow-up.


**Recommendation** **5:**

**The Cystic Fibrosis Foundation recommends conducting *CFTR* variant screening twice weekly or more frequently as resources allow.**


The timeliness of CF NBS is influenced by a multitude of factors, including those related to the testing laboratory, such as shipping methods and frequency, laboratory hours, and variable testing frequency of dried bloodspots [72,73,74,75]. Screening *CFTR* variants at least twice per week rather than once a week was associated with an earlier age of diagnosis [72]. Newborn screening programs should perform variant testing at least twice weekly and should also consider evaluating the other time-related factors (such as second specimen collections, sample shipping and processing, batch size, and staffing) that enable and support testing at this frequency or more often.


**Recommendation** **6:**

**The Cystic Fibrosis Foundation recommends the inclusion of a *CFTR* sequencing tier following IRT and *CFTR* variant panel testing to improve the specificity and positive predictive value of CF newborn screening.**


A growing number of NBS programs have employed third-tier *CFTR* sequencing (SEQ), also called extended gene analysis (EGA), within the CF NBS algorithm, in which infants with one CF-causing variant following panel testing undergo analysis of all exons, flanking introns, and specific intronic variants (Figure 1). Typically, these programs only refer infants with two CF-causing *CFTR* variants for sweat testing, though this may vary by jurisdiction. It should be noted that sequencing results may include variants not yet interpreted by CFTR2, but which are classified by testing laboratories using the ACMG variant interpretation criteria [11]. Variants classified as pathogenic or likely pathogenic using these criteria will prompt the same follow-up as variants classified as CF-causing by CFTR2. The *CFTR* sequencing tier was first introduced in California to minimize false positive results, reduce the number of sweat tests, and improve detection in a genetically heterogeneous population [9].

The sensitivity of CF NBS programs utilizing a *CFTR* sequencing tier was generally >90% and comparable to some programs utilizing IRT-DNA; however, specificity and PPV were significantly improved by inclusion of sequencing, ranging from 99% to 100% and 34–77%, respectively [9,49,76,77]. A study in California modeling IRT-DNA-SEQ protocols found that the maximum CF case detection of 97% was achieved using a California-specific DNA panel of 40 variants or a CFTR2-based DNA panel of over 200 CF-causing variants, followed by sequencing [49]. Notably, this study demonstrated profound improvement in the detection of infants who were Asian, Black, and Hispanic when third-tier sequencing was utilized; detection for these groups with a second-tier panel ranged from 17% to 79% but improved to 83–100% with third-tier sequencing.

For all programs, the utility of *CFTR* sequencing and its impact on sensitivity, specificity, and PPV are also dependent on the IRT threshold and size of the second-tier DNA panel (discussed above in **Recommendations 1 and 4**), as infants will not progress to the sequencing tier if not identified during the first two tiers of the NBS algorithm.

*CFTR* sequencing will not detect all variants. Structural variants and deep intronic variants, some of which have higher frequencies in specific ancestral groups, will not be detected unless deletion/duplication testing is performed or they are specifically targeted. Currently, genomic-level *CFTR* sequencing (including all introns and up- and down-stream regions) is not being performed as part of CF NBS but may be incorporated in the future to overcome some of these limitations.

The decision of whether to include a *CFTR* sequencing tier in a CF NBS algorithm should be made in the context of the goals and current performance of the NBS program. Reduction in sweat testing of carriers can be achieved if only infants with two CF-causing, pathogenic, or likely pathogenic *CFTR* variants are considered screen-positive and referred for follow-up (Figure 2; see also *Considerations for Implementation*). Some programs may also choose to include variants of varying clinical consequences (VVCC) or variants of uncertain significance (VUS) based on their program goals related to increasing the detection of CRMS/CFSPID. The parents of infants with only one variant detected after sequencing (presumed carriers) should be notified and recommended to receive genetic counseling, as previously noted in the discussion following **Recommendation 4**. Population shifts should be considered, and program sensitivity should be periodically re-evaluated.

### 3.3. Communication Following Newborn Screening


**Recommendation** **7:**

**The Cystic Fibrosis Foundation recommends that both the primary care provider and the CF specialist be notified of abnormal newborn screening results.**


Once CF NBS laboratories have a positive result, a medical provider is notified. The medical provider is responsible for disclosing the NBS results to the family, assessing the infant’s immediate condition, and arranging the diagnostic evaluation and clinical follow-up. Newborn screening programs in the US vary in notification practices: some notify just the primary care provider (PCP), some notify just the CF specialist, and some notify both the CF specialist and PCP.

Infants with a positive CF NBS identified by programs that notified both the PCP and the CF specialist had an earlier first visit than when only the PCP was notified, and a delayed diagnostic evaluation was less likely [78,79,80]. Families preferred to have a CF specialist notified of the positive NBS results and wanted prompt contact from a health care provider who is informed and knowledgeable about CF and their infant’s risk for CF; this allows timely follow-up to minimize emotional distress while awaiting diagnosis [78,80,81,82,83,84,85,86]. There is also evidence that families want to receive information on their infant’s NBS from a trusted medical provider they know, such as their PCP [84,87,88]. However, infants from American Indian, Asian, Black, Hispanic, or multiracial populations, whose results are only communicated to the PCP, may have delayed referral after a positive NBS due to the misguided perception in the medical field that CF only affects non-Hispanic White infants [31]. Notifying both the PCP and the CF specialist increases the likelihood of a correct assessment of the infant’s CF risk and that accurate information about CF is shared with families [87,88].

It is essential to have a well-defined, consistent, and reproducible system to ensure that each infant with a positive NBS result receives follow-up and treatment in a timely manner. If the initial contact is with a PCP, the PCP should be contacted by telephone, as this was associated with earlier sweat chloride concentration testing [89]. There must be a clear delineation of responsibility for notification of families and scheduling follow-up to prevent delays or miscommunication [90], including when incorrect PCP information is provided [91]. Educational materials and report recommendations should be standardized and include up-to-date and easy-to-understand information with the next steps for PCPs and families.

## 4. Discussion

### 4.1. Considerations for Implementation

The goals of a CF NBS program should be well-defined and address topics such as sensitivity, specificity, and equity. Programs must also determine their goals related to the detection of carriers and infants with CRMS/CFSPID, which is not the primary purpose of CF NBS but a byproduct that depends on the algorithm used and has considerable implications for families and the support and resources required for follow-up. A flowchart for programs to consult related to the implementation of the Consensus Recommendations is provided in Figure 2, with indications of where program goals should dictate decision-making.

Further considerations related to implementation are available in Table 3 (ways to improve equity in CF NBS) and are written below. It should be noted that these considerations were written by the authors and agreed upon by the committee but not formally evaluated in a systematic review or voted upon to achieve consensus, and they do not represent formal CF Foundation recommendations.

Programs should review the sensitivity of their CF NBS overall and by races, ethnicities, and subpopulations (for example, Hmong, Cambodian, or Ecuadorian). Systems should be in place to report false-negative NBS results and to review the reasons for false-negative NBS results, including IRT thresholds and *CFTR* variants in missed cases.NBS programs should consider IRT threshold changes and the makeup of the *CFTR* variant panel based on the evaluation of false-negative cases, with particular scrutiny of missed infants of historically marginalized races or ethnicities [48,69,70].Rigorous systems that track outcomes and involve CF specialists, PCPs, and NBS programs can ensure both short- and long-term follow-up of infants with CF, including those with true-positive and false-negative NBS [92]. Programs should specifically track people with CF who had a delayed diagnosis or a false-negative NBS [9] and use the data for continuous improvement in all steps of the NBS protocol.NBS programs using a more limited second-tier *CFTR* variant panel should consider using the VHIRT strategy and include an unequivocal and obvious alert in the NBS report that a risk of CF may be present, especially if there is a family history of CF or if the infant has signs and symptoms of CF, including but not restricted to persistent diarrhea, poor weight gain, salt loss syndrome, chronic cough, or respiratory problems. Examples of resources that newborn screening programs can use are the ACMG ACT sheets, which can be found on the ACMG website: (https://www.acmg.net/ACMG/Medical-Genetics-Practice-Resources/ACT_Sheets_and_Algorithms.aspx [accessed on 5 February 2025]).NBS programs should exercise caution when utilizing a VHIRT strategy for infants in the NICU, as IRT is routinely elevated in premature infants or with perinatal stress. The Clinical and Laboratory Standards Institute (CLSI) guideline regarding NBS for infants in the NICU should be consulted for further information [93].Education should be provided to all medical providers, emphasizing that CF NBS is a *screening* test, not a diagnostic test, and will not detect all cases of CF. Sweat testing and diagnostic follow-up should be considered for any infant, child, or adult with clinical symptoms suggestive of CF or a family history of CF, even if that individual’s CF NBS was normal.Programs with the goal of increasing the specificity and positive predictive value of CF NBS, thereby reducing the number of sweat tests performed on infants who are likely CF carriers or who have CRMS/CFSPID, should identify infants as having a positive CF NBS only when *two* CF-causing, pathogenic, or likely pathogenic *CFTR* variants are detected by third-tier sequencing. This approach increases the risk of false-negative tests, as structural and deep intronic *CFTR* variants may be missed with this strategy; some sub-populations in which specific exon deletions or intronic variants are common may have a disproportionately higher false-negative risk.NBS programs that screen for all CF-causing variants in CFTR2, but do not implement a *CFTR* sequencing tier, may consider referring only infants with *two* CF-causing *CFTR* variants for sweat testing and diagnostic follow-up if this achieves 95% sensitivity and aligns with program goals.If the goals of NBS programs include identifying all infants at risk for CF or CFTR-related disorders (CFTR-RD), NBS programs should refer all infants with elevated IRT and only one CF-causing *CFTR* variant identified for diagnostic follow-up. This will lead to increased detection of infants with CRMS/CFSPID. Jurisdictions should ensure that CF centers have adequate sweat testing capability, clinical staff, and access to CF-specific genetic counseling for timely follow-up.Per the CF Foundation-endorsed guideline, *Genetic counseling access for parents of newborns who screen positive for cystic fibrosis: Consensus guidelines*, parents of all infants with a positive CF NBS result should be offered genetic counseling by a provider with expertise in CF and training in genetic counseling [38].Programs referring only infants with two *CFTR* variants for sweat testing and diagnostic follow-up should continue to report presumed carrier status (only one CF-causing *CFTR* variant identified) with a referral for genetic counseling. However, programs have the discretion to recommend that this be coordinated via the infant’s primary care provider, the state health department, or a CF care center.At the time of this publication, there are currently no commercially available *CFTR* variant panels that include all CF-causing variants in CFTR2. Expanding to a *CFTR* variant panel containing several hundred variants will necessitate a switch to next-generation sequencing if not already in use. Programs can consider using external referral labs or partnering with other states to implement next-generation sequencing.Infants whose parents are known CF carriers or infants with a full sibling with CF should have a diagnostic work-up for CF regardless of the CF NBS result. If the NBS is normal and prenatal genetic testing indicates that the infant has fewer than the two *CFTR* variants identified in the parents, then providers can consider not testing for CF.Infants with CF born to a mother or birthing parent taking a CFTR modulator therapy during pregnancy may have a false-negative NBS due to low IRT that occurs with fetal exposure to the modulator [94]. Newborn screening programs should have open lines of communication with CF clinicians, who will likely identify these infants and may ask the NBS program to consider DNA testing regardless of, or instead of, IRT measurement to reduce the risk of a false-negative result [95,96]. The risk of CF should be carefully evaluated in these infants, and diagnostic testing should be considered.Infants with CF and meconium ileus may have a false-negative NBS due to a low IRT and should be referred for diagnostic evaluation. Some CF NBS programs perform DNA testing for all babies with meconium ileus, regardless of IRT value, to expedite diagnoses for those with CF.

It is recognized that the feasibility of implementing these Consensus Recommendations will vary by state, depending on both the resources available and the degree of change required from the CF NBS algorithm currently being used. This guideline intends to provide guidance that improves outcomes for infants with CF by improving equity and timeliness of CF diagnosis, and it is expected that states will continually evaluate processes and outcomes to assess the impact of the guidelines and share implementation strategies. We did not consider or address costs, but we recognize their significant influence on implementation. States are encouraged to make incremental changes to improve their CF NBS protocols as they work toward full guideline implementation.

**Table 3 IJNS-11-00024-t003:** Improving cystic fibrosis newborn screening: benefits and impact on equity.

Newborn Screening Step	Best Practices for Newborn Screening Programs to Benefit All Infants
Immunoreactive Trypsinogen (IRT) Level	Choose a floating IRT level, as IRT levels may vary by race and ethnicity, season, and reagent lot [48,97]
	Inclusion of a very high IRT (VHIRT) strategy reduces false-negative NBS in infants who have rare variants not included on *CFTR* variant panels, and thus have false-negative NBS, which is more common in infants who are American Indian, Asian, Black, Hispanic, or multiracial.
DNA Testing	Screen for all CF-causing *CFTR* variants in CFTR2 to reduce false-negative NBS in infants with rare variants, which is more common in infants who are American Indian, Asian, Black, Hispanic, or multiracial. This increases the percentage of infants who will have two *CFTR* variants detected, which shortens the time to diagnosis and initiation of treatment.
	Evaluate the performance of variant panel sensitivity overall and by race and ethnicity to ensure adequate detection. Consider adding *CFTR* variants from false-negative cases to the screening panel to increase sensitivity.
	Consider adding third-tier *CFTR* sequencing to increase the detection of infants with rare variants, which is more common in infants who are American Indian, Asian, Black, Hispanic, or multiracial.
Communication	Notification of a positive NBS should involve the primary care provider, as there is likely more trust built with the family, which is particularly important in families of historically marginalized and underserved populations who may have experienced discrimination and trauma from the health care system. Notification of a positive NBS should also involve the CF specialist, who can correctly convey that CF occurs in infants of all races, ethnicities, and ancestries and arrange for rapid diagnostic testing.
	Information about NBS results should be given to families in their native language in both oral and written form and at a level that they can understand. Certified interpreters should be used if the provider is not proficient in a family’s language. A health literacy universal precautions approach should ensure that information is comprehensible to those with low or varying degrees of health literacy [98,99].
	Newborn screening programs and CF specialists should have rigorous systems in place to track all positive NBS results and infants with false-negative screens. Systems should ensure that all infants can access diagnostic work-up, timely referrals, and early treatment. Barriers to timely diagnosis (by 4 weeks of life) and care should routinely be assessed, identified, and addressed [34]. Reasons for false-negative screens (e.g., IRT below threshold, variants not detected) should be tracked and used to improve the NBS program.
Education of the Medical Community	The medical community should be educated that NBS is a screening test only and that all people with signs or symptoms of CF should have a diagnostic evaluation, regardless of the NBS result.
	The medical community should be educated that CF occurs in people of all races, ethnicities, and ancestries.

### 4.2. Future Research Considerations

We developed this guideline for CF NBS based on the best available evidence supported by rigorous systematic reviews. Nevertheless, recommendations are made by expert consensus due to limited high-quality evidence. Dissemination of program outcomes and publishing work that improves NBS outcomes is not the primary function of public health programs, which are often poorly resourced, likely contributing to the paucity of literature. Further research and peer-reviewed publications are needed to improve CF NBS timeliness, sensitivity, and equity. Moreover, additional published evidence is essential to guide the future development of standards for CF NBS in the US, ensuring that screening programs have clear benchmarks to meet.

The CF NBS Guidelines Committee identified the following as the most pressing research questions in CF NBS: (1) What are the best practices to improve communication among the NBS program, PCP, and CF specialist; (2) How can culturally aware communication with families improve timeliness and follow-up; (3) How can the detection of *CFTR* variants be improved in ancestral populations with rare and/or understudied *CFTR* variants (e.g., Asian and African populations) [16]; (4) What measures exist to correct the misbelief in the medical field that CF only exists in the non-Hispanic white population; (5) How do extreme weather conditions or natural disasters impact NBS laboratory processing and subsequent timeliness of infant evaluation; (6) What are best practices to screen and diagnose infants for whom the typical CF NBS process may not occur as intended, such as infants in remote rural areas, infants born in other countries due to parental military deployment or other employment, infants using tribal health services, and infants in the neonatal intensive care unit; and (7) How does regionalization of CF NBS (i.e., one NBS lab providing services for multiple states) impact sensitivity, equity, and timeliness?

## 5. Conclusions

Newborn screening programs have allowed for early diagnosis, improved monitoring, and early treatment for most infants with CF. It is crucial for programs to engage in continued process improvement to ensure that all infants with CF benefit from timely and accurate identification. In the US, variations among states evolved as relatively rapid implementation occurred between 2005 and 2010. Better knowledge and significant technological advances have made it possible to apply quality improvement and best practices nationwide. The CFF is committed to assisting states in achieving improved NBS algorithms and will accompany this process and product with advocacy efforts.

## Figures and Tables

**Figure 1 IJNS-11-00024-f001:**
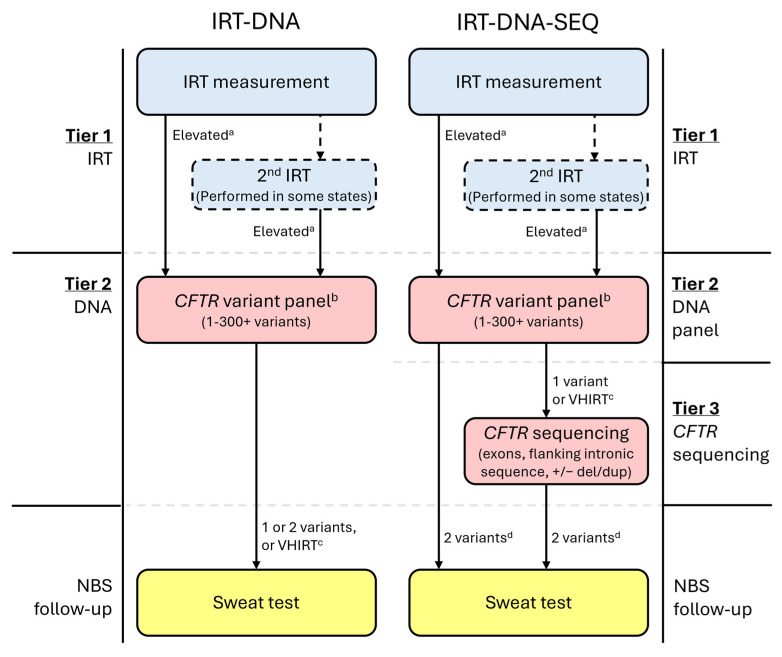
Example CF newborn screening algorithms for protocols using either IRT-DNA or IRT-DNA-SEQ. ^a^ State-determined; may be fixed or floating. ^b^ Variant panels may use sequencing methodology; however, if the list of variants to be released/reported is limited and pre-determined, then this is still considered a variant panel. ^c^ Not all states will utilize a very-high IRT (VHIRT) referral strategy. ^d^ Most states performing *CFTR* sequencing will only refer infants with two CF-causing variants for sweat testing.

**Figure 2 IJNS-11-00024-f002:**
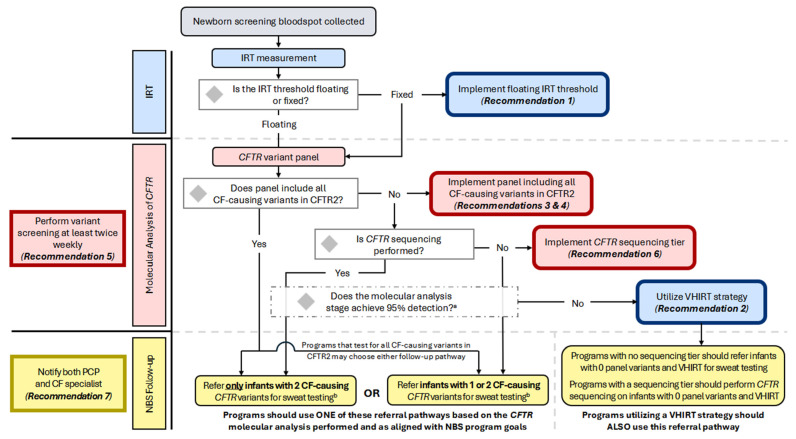
Flowchart of the CF NBS algorithm and decision points for implementing consensus recommendations. All US programs initiate the CF NBS algorithm by measuring IRT; some perform two IRT measurements. Programs using a fixed IRT threshold should implement a floating IRT cutoff. Infants progressing to molecular analysis will undergo *CFTR* variant panel testing. This may be performed using sequencing technology if results are limited to a pre-determined list. Programs whose panel includes all CF-causing *CFTR* variants may choose whether to refer for sweat testing only infants with two *CFTR* variants OR infants with one or two *CFTR* variants. A program’s choice to refer only infants with two variants will likely result in decreased detection of CRMS/CFSPID, decreased sweat testing of carriers, but an increased risk of missing CF-affected infants with rare variants. The choice to refer infants with one or two variants will likely result in increased CRMS/CFSPID detection, increased sweat testing of carriers, but improved detection of CF-affected infants with rare variants. Programs whose variant panel does not include all CF- causing variants in CFTR2 and who are not performing *CFTR* sequencing should continue to refer infants with one or two variants detected. Programs performing *CFTR* sequencing following variant panel testing should refer only infants with two CF-causing variants. All programs should assess the sensitivity of their molecular analysis stage; if 95% detection within major ancestral sub-populations is not achieved and the variant panel does not include CF-causing variants from CFTR2, programs should maintain or implement a very-high IRT (VHIRT) referral strategy. ^a^ It is recommended that a variant panel achieve 95% detection rate in major ancestral sub-populations; ^b^ When making the determination of which infants should be referred for sweat testing, some programs will choose to include and report variants with interpretations other than CF-causing, such as variants of varying clinical consequences (VVCCs), variants of uncertain significance (VUS), and pathogenic or likely pathogenic variants.

**Table 1 IJNS-11-00024-t001:** Definitions.

Term	Definition
CF Newborn Screening Process Terms	
Immunoreactive trypsinogen (IRT)	A pancreatic enzyme precursor that is found in the blood. Measurement of IRT is the first step in the CF NBS process in all US states, and IRT is usually elevated in infants with CF. IRT may be elevated in some CF carriers or for reasons unrelated to CF (traumatic birth, premature birth, ancestral background, ambient temperature, nutritional status, or other reasons).
DNA panel	A genetic test that determines the presence or absence of a limited list of *CFTR* variants. Panel testing will evaluate only for variants on a pre-determined list, typically comprising recognized CF-causing variants. Panels may vary in size from 23 to >300 variants, and their sensitivity varies by the ancestry of the person being tested. A DNA variant not included on the panel will not be evaluated or reported if present. DNA panel testing may be performed using sequencing technology if only a limited and pre-determined list of variants are reported.
*CFTR* sequencing	A genetic test that determines the presence, absence, and order of nucleotides in *CFTR* DNA. Sequencing may be performed using a variety of methods and may include the entire *CFTR* gene or limited portions only, such as the coding and flanking regions (exons and a limited number of intronic nucleotides near intron-exon junctions). In the context of NBS, most *CFTR* sequencing tests evaluate only the exons and flanking intronic regions, with limited analysis of specific deep intronic variants (see below for definitions of terms).
**CF Newborn Screening Evaluation Measures**	
Sensitivity	The ability for a test to identify people with a condition as having positive/abnormal results. A test with high sensitivity has very few false-negative results; it is rare for a person with the condition to be “missed” or given negative results. Tests with high sensitivity will identify almost every person who actually has the condition of interest.
Specificity	The ability for a test to identify people who do not have a condition as having negative/normal results. A test with high specificity has very few false-positive results; it is rare for a person who does not have the condition to have a test result that is positive/abnormal.
Positive predictive value (PPV)	The probability that an individual with a positive/abnormal test result actually has the condition of interest. Tests with high PPV have few false-positive results because almost everyone that the test “catches” does have the condition.
False-negative result	A negative or normal CF NBS result in an infant later confirmed to have CF. These infants may be described as having been “missed by newborn screening.”
False-positive result	A positive or abnormal CF NBS result in an infant who does NOT have CF.
**Genetic Testing Terms**	
Exon	A segment of DNA that codes for a protein; exons are often referred to as the “coding region” and are always included in a clinical *CFTR* sequencing test.
Flanking region	A limited number of intronic DNA base pairs (often 10–50) that lie immediately adjacent to the exons; these small portions of the intron are typically included in a clinical *CFTR* sequencing test.
Intron	A segment of DNA that does not code for a protein; introns are often referred to as the “non-coding region” and may not be included in a clinical *CFTR* sequencing test unless specified.
Full-gene *CFTR* sequencing	A sequencing test that looks at the entire *CFTR* gene and includes all exons and all portions of all introns. This is a new test that is different than the typical *CFTR* sequencing test that has been used over the past decades. This test may also be called whole-gene *CFTR* sequencing and typically uses next-generation sequencing technology. In the context of NBS, this type of testing is not currently being used.
Deletion/duplication testing	An evaluation for large structural variants that typically include one or more exons (e.g., CFTRdele2,3, which is a deletion of exons 2 and 3 in *CFTR*). Large deletions and duplications involving exons cannot be detected by Sanger sequencing and may require an additional test, such as multiplex ligation-dependent probe amplification (MLPA). Some next-generation sequencing tests can detect large deletions and duplications involving exons, but additional analysis might be needed. It is important to note whether a *CFTR* sequencing test will include deletion/duplication analysis.
**Types of *CFTR* Variants**	
Variant interpretation	A prediction of the molecular and/or phenotypic consequence of a given DNA variant. Variant interpretation definitions according to CFTR2 (https://cftr2.org) are provided in this table and used throughout the manuscript. Many clinical testing laboratories will also use variant interpretation definitions recommended by the American College of Medical Genetics and Genomics [11].
CF-causing variant	A *CFTR* variant that is expected to cause CF when found in *trans* with another CF-causing variant. Laboratories may use the terms “pathogenic” or “likely pathogenic” [11] when describing CF-causing variants or those expected to cause CF, but which are not interpreted by CFTR2. In this manuscript, CF-causing variants as defined by CFTR2 are recommended for assessment and reporting as part of CF NBS. It is recognized that not all CF-causing variants are readily detectable by sequencing technology. Therefore, the NBS program should focus on CF-causing variants interpreted as such by CFTR2 that are located in the coding and flanking regions of *CFTR*. Programs may further choose whether to include structural variants or deep intronic CF-causing variants, as program technology and resources allow.
Non-CF-causing variant	A *CFTR* variant that is NOT expected to cause CF, even when found in *trans* with a CF-causing variant. Some individuals with CF have non-CF-causing variants, but it is expected that these variants are not the cause of the disease. Non-CF-causing variants may be present in addition to two CF-causing variants. Laboratories may use the terms “benign” or “likely benign” [11] when describing non-CF-causing variants or *CFTR* variants not yet interpreted by CFTR2, but which are known to have no clinical significance.
Variant of varying clinical consequences (VVCC)	A *CFTR* variant that may result in CF in some people but not in others, when found in *trans* with a CF-causing variant. Individuals with VVCCs who do not have CF may have a *CFTR*-related disorder or no symptoms. Individuals with CRMS/CFSPID frequently have at least one VVCC [12].
Variant of uncertain significance (VUS)	A *CFTR* variant for which the clinical significance is not clear. Over time and as more data are collected, some VUS may be re-interpreted as CF-causing, non-CF-causing, or VVCCs. Clinicians are recommended to review VUS regularly to see if the interpretation has changed.

Adapted with permission from [13].

**Table 2 IJNS-11-00024-t002:** Consensus Recommendation Statements.

Recommendation		Percentage Agreement
Immunoreactive Trypsinogen (IRT)		
1	The Cystic Fibrosis Foundation recommends the use of a floating immunoreactive trypsinogen cutoff over a fixed immunoreactive trypsinogen cutoff.	100%
2	The Cystic Fibrosis Foundation recommends using a very high immunoreactive trypsinogen referral strategy in CF newborn screening programs whose variant panel does not include all CF-causing variants in CFTR2 or does not have a variant panel that achieves at least 95% sensitivity in all ancestral groups within the state.	100%
***CFTR* Variant Testing**		
3	The Cystic Fibrosis Foundation recommends that CF newborn screening algorithms should not limit *CFTR* variant detection to the F508del variant or variants included in the ACMG-23 panel.	100%
4	The Cystic Fibrosis Foundation recommends that CF newborn screening programs screen for all CF-causing *CFTR* variants as identified by CFTR2.	100%
5	The Cystic Fibrosis Foundation recommends conducting *CFTR* variant screening twice weekly or more frequently as resources allow.	100%
***CFTR* Sequencing**		
6	The Cystic Fibrosis Foundation recommends the inclusion of a *CFTR* sequencing tier following IRT and *CFTR* variant panel testing to improve the specificity and positive predictive value of CF newborn screening.	100%
**Communication**		
7	The Cystic Fibrosis Foundation recommends that both the primary care provider and the CF specialist be notified of abnormal newborn screening results.	100%

## Data Availability

Full details of the systematic reviews, including spreadsheets for each PICO question with a list of reviewed manuscripts, review metrics, and inclusion/exclusion status, are available upon request from the corresponding author. Please contact Meghan McGarry at meghan.mcgarry@seattlechildrens.org.

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
