# Peer review of "Cystic Fibrosis Newborn Screening: A Systematic Review-Driven Consensus Guideline from the United States Cystic Fibrosis Foundation"

_2409-515X, 2025, doi:10.3390/ijns11020024_

Round 1

Reviewer 1 Report

Comments and Suggestions for Authors

The authors have produced CFF (US) guidance on the performance of newborn bloodspot screening for CF.   A timely piece of work, the recommendations are clear and pertinent.  It is particularly encouraging to see statements that consider strategies to address potential inequity across ethnic populations, most notably the recommendation to consider a VHIRT strategy if variant coverage is inadequate for the screened population and recommendations with respect to processing of results, which should improve timeliness for all populations.

Overall, the paper is well written and the approach clear.

I have some questions, which the authors should be able to address.

  1. Six systematic reviews were completed, but no details are included. Can the methodology be included as a supplement, especially the search strategy and quality appraisal tool.  In the discussion (lines 538-544), the authors describe a paucity of high quality evidence.  This is fine (and not unexpected) but it is important not to imply that the seven statements are generated from high quality research papers.  Undertaking a systematic review isn’t in itself a guarantee of clear evidence and this should be outlined clearly in the results (caveats).  Can the reviews be included as a supplement and important methodological details included as described above.
  2. That said, the seven statements are excellent and I believe will impact positively on practice. They are universally agreed on (100%), but again some details of the methodology with regards to this consensus process is needed (how many people involved, who (stakeholders?), what method (online or F-2-F), delphi consensus?).
  3. The authors “flesh out” these seven statements with a further fifteen bullet points in the discussion, some expanding on the previous, some raising separate issues. A theme that seems to emerge from this section is that different states appear to have different objectives with respect the overall aim of the CF programme, for example carrier reporting and the recognition of infants with CRMS/CFSPID or even a CFTR-related disorder.  Isn’t it reasonable to have, as a starting point, an agreement across all states as to the aims of NBS for CF?  I can imagine that this is challenging, but at a minimum the authors should acknowledge this fundamental issue in achieving a co-ordinated approach.  Achieving agreed aims and standards for outcome metrics (sensitivity and PPV especially) should be listed with the further research questions.
  4. A bullet point raises the important issue of mothers taking CFTR modulator therapy. The statement is somewhat reassuring with respect to lines of communication, but the reality is quite complex and concerning.  Mothers (CF and sometimes carriers) work with their adult CF team and obstetrician through the pregnancy (hopefully in a partnership manner) to manage risk.  These adult physicians may not always be aware of the implications of maintaining CFTRm therapy through pregnancy on NBS outcomes and sometimes do not have close contact with adjacent Paediatric services, so work is needed to disseminate this knowledge.  It would be helpful to be more explicit on the challenges of joined up thinking across these services.
  5. With respect to the recommendation of using a floating IRT cut off, I think it would be helpful to include some additional practical advice to help labs achieve this goal. In the UK labs felt they were “chasing their tails” with batch variation and have opted to return to a fixed cut-off (in reality our cut-offs have stayed stable for years, like our weather).  I think a floating cut-off is a better approach, but it is important to acknowledge the challenge of managing this (buddy groups etc). 
  6. Also it is important to ensure labs have enough data before changing cut offs (ie, not undertaking a knee jerk reduction in IRT-1 following a couple of false negative results). In the ECFS survey/guidance (ref 71) we describe some modelling to assess the reliability of sensitivity measurements and suggest caution in interpreting this measure for programmes recognising less than 40 infants with CF/year.
  7. With respect to the VHIRT strategy, I think this is an excellent section, but can the authors clarify that the 95% variant coverage is for all populations and not an overall measure, for example a programme may have an overall variant coverage of 96%, but only 90% or less for the non-white population. Would the VHIRT strategy be recommended in this case? (figure 2 b)
  8. The VHIRT strategy is important to improve equity, but it is not without issues (poor PPV) and these are outlined in the manuscript, so I agree with the assertion that with sufficient variant cover (all CFTR2 characterised CF causing variants), the VHIRT strategy can be abandoned, again it would be useful to clarify that all populations should be assessed for variant cover. If programmes do switch to EGA, careful assessment of the impact on sensitivity for all populations is necessary (and other metrics).
  9. With respect to screening for all CFTR2 CF causing variants, this is a good recommendation but then the authors state that VVCCs can be included depending on the states approach to CRMS/CFSPID recognition (line 317). This relates to my comment in 3), and is not consistent with the rest of the paper.  Research should determine clear agreed goals for the programme.   The authors should be explicit that including VVCCs or even unrestricted extensive genetic analysis with VUS’s will significantly increase CRMS/CFSPID recognition.     
  10. With respect to processing NBS results, this may be the most impactful recommendation to address some of the timeliness issues reported from states. I think from a global perspective it would be useful to clarify the relationship between the PCP and the CF team in the US.  For example, in the UK, whilst most GPs are excellent, it would be impossible to guarantee a consistent and appropriate approach for families.  I think the situation may be different in the US, with PCPs more atuned to screening and better placed to organise sweat testing.  Would the CF team actively contact the PCP or wait to hear from them?  How does this relationship work (I suspect it varies from state to state)?
  11. A statement that these recommendations have global significance would be helpful (most molecular genetics labs in the UK have moved to weekly testing to save money).
  12. The research questions are all pertinent, but I think the authors should reflect on how they facilitate the adoption of these excellent standards across the US (and further). I think there is too much wriggle room in the paper and programmes are likely to continue the status quo.  An important research aim should be to establish standards that include timeliness, PPV, sensitivity, carrier recognition and CRMS/CFSPID approach.  With these metrics, under-performing programmes will need to translate these recommendations into practice.
  13. Small comment, in Europe we use the term “safety net” not “failsafe” for VHIRT strategy
  14. Tables and Figures are clear
  15. References are numbered twice

Reviewer 2 Report

Comments and Suggestions for Authors

Very interesting paper.

Few clarifications would be helpful.

  1. Precise what do you mean by timely diagnosis
  2. How does the time of diagnosis differ in different ethnic groups compared to the white/Caucasian race?

  3. What IRT cut-off is applied for the EGA (SEQ) protocol?"
  4. From global NBS programs perspective cut-offs of 99,4 percentile are pretty common , not considered VHIRT

  5. v.494-498 In European NBS programs, CFSPIDs are undesirable, and it is from them that CFTR-RDs are recruited
  6. v.519-525 

    Instead of 'parent,' it might be more precise to specify 'mother,' as if the father takes modulators, the fetus is not exposed to their effects. The information about a sick father should definitely reach the laboratory

  7. v.526-528 Could you provide specific references?

Reviewer 3 Report

Comments and Suggestions for Authors

I am thanking the authors for providing an interesting and very important document -  systematic review-driven consensus guideline for Cystic Fibrosis Newborn Screening throughout the United States. The manuscript is very informative, well-written and is fully supported by existing literature. Yours consensus recommendations are intended to be location-agnostic and are applicable to newborns screened for CF throughout the US. In contrast to other CFF guidelines, this guideline has recommendations pertinent to public health practice.
I have few minor comments.
1)    Page 3/22, first line of the Table 1. It is written: … or for reasons unrelated to CF (traumatic birth, premature birth, ancestral background, ambient temperature, nutritional status). Are you sure that you have mentioned all the possible reasons? Wouldn’t it be better to add at the end of the sentence ….. ambient temperature, nutritional status or for some other reason.).?
2)    Line 335 and Line 337 – CRMS – I think you need to write CRMS/CFSPID as it is mentioned in Table 1?
3)    Line 465-469 – It is written: “NBS programs using a more limited second-tier CFTR variant panel should consider using the VHIRT strategy and include an unequivocal and obvious alert in the NBS report that a risk of CF may be present, especially if there is a family history of CF or if the infant has signs and symptoms of CF, such as persistent diarrhea, poor weight gain, chronic cough, or respiratory problems”. To my mind it is very important to mention at least another one symptom – salt syndrome (Pseudo-Bartter syndrome). And wouldn’t it better to write – “… symptoms of CF, including, but not restricted to persistent diarrhea, poor weight gain, salt loss syndrome, chronic cough, or respiratory problems.” 

Reviewer 4 Report

Comments and Suggestions for Authors

Dear Authors

The manuscript is very well written and highly relevant.

The guidelines and data behind the guidelines are clear and well described.

I have very few comments to manuscrip.

  1. If possible, it would be nice to have some data on median timeline from receival of the NBS sample to the screen positives CF screening result and the final CF diagnosis.
  2. Is there a recommended level of specificity equivalent to that of sensitivity? Would adding of sequencing increase the number of CRMS and CFSPID cases and does CRMS and CFSPID cases count as true positives? 

Round 2

Reviewer 1 Report

Comments and Suggestions for Authors

The authors have addressed most of my comments.

It is a shame they haven't provided further information on managing a floating cut-off as they feel this outside the scope of their work.  I get that, but this is just the type of helpful detail that can drive forward standards and to me, the IJNS seems an appropriate screening specific journal in which to include such detail.

But otherwise OK (not so sure about a consensus process in a room of 14, but at least it clear now!) and well done again for producing such a useful piece of work

Reviewer 3 Report

Comments and Suggestions for Authors

An interesting and very important document -  systematic review-driven consensus guideline for Cystic Fibrosis Newborn Screening throughout the United States. The manuscript is very informative, well-written and is fully supported by existing literature. Yours consensus recommendations are intended to be location-agnostic and are applicable to newborns screened for CF throughout the US. In contrast to other CFF guidelines, this guideline has recommendations pertinent to public health practice.

I am absolutely satisfied with all the changes.